# Cross-species analysis defines the conservation of anatomically segregated VMH neuron populations

**Alison H Affinati[1], Paul V Sabatini[1], Cadence True[2], Abigail J Tomlinson[1], Melissa Kirigiti[2], Sarah R Lindsley[2], Chien Li[3], David P Olson[4], Paul Kievit[2], Martin G Myers[1]\*, Alan C Rupp[1]\***

[1]Department of Internal Medicine, University of Michigan, Ann Arbor, United States; [2]Oregon National Primate Research Center, Beaverton, United States; [3]Novo Nordisk Research Center, Seattle, United States; [4]Department of Pediatrics, University of Michigan, Ann Arbor, United States

**Abstract** The ventromedial hypothalamic nucleus (VMH) controls diverse behaviors and physiologic functions, suggesting the existence of multiple VMH neural subtypes with distinct functions. Combing translating ribosome affinity purification with RNA-sequencing (TRAP-seq) data with single-nucleus RNA-sequencing (snRNA-seq) data, we identified 24 mouse VMH neuron clusters. Further analysis, including snRNA-seq data from macaque tissue, defined a more tractable VMH parceling scheme consisting of six major genetically and anatomically differentiated VMH neuron classes with good cross-species conservation. In addition to two major ventrolateral classes, we identified three distinct classes of dorsomedial VMH neurons. Consistent with previously suggested unique roles for leptin receptor (*Lepr*)-expressing VMH neurons, *Lepr* expression marked a single dorsomedial class. We also identified a class of glutamatergic VMH neurons that resides in the tuberal region, anterolateral to the neuroanatomical core of the VMH. This atlas of conserved VMH neuron populations provides an unbiased starting point for the analysis of VMH circuitry and function.

**\*For correspondence:**
mgmyers@umich.edu (MGM);
ruppa@med.umich.edu (ACR)

## Introduction

The ventromedial hypothalamic nucleus (VMH, which primarily contains glutamatergic neurons) plays important roles in a variety of metabolic responses and in the control of behaviors relevant to panic, reproduction, and aggression. The VMH contains several anatomic subdivisions, including the dorso-medial and central VMH (VMH$_{DM}$ and VMH$_C$, respectively, which control autonomic outputs and behavioral responses to emergencies; *Lindberg et al., 2013*; *Vander Tuig et al., 1982*), and the ventrolateral VMH (VMH$_{VL}$; known for roles in sexual and social behaviors; *Hashikawa et al., 2017*; *Krause and Ingraham, 2017*). The predominantly GABAergic tuberal region of the hypothalamus lies anterolateral to the core of the VMH.

Each VMH subdivision mediates a variety of outputs and thus presumably contains multiple functionally distinct cell types. For example, activating adult *Nr5a1*-expressing VMH neurons (which includes most cells in the VMH$_{DM}$ and VMH$_C$) promotes panic-related behaviors, augments hepatic glucose output to increase blood glucose, and elevates energy expenditure (*Meek et al., 2016*, 201; *Flak et al., 2020*; *Kunwar et al., 2015*). In contrast, activating the subset of VMH$_{DM}$ cells that expresses leptin receptor (*Lepr*, which encodes the receptor for the adipose-derived, energy bal-ance-controlling hormone, leptin; *Chen et al., 1996*; *Tartaglia et al., 1995*) promotes energy expen-diture without altering these other parameters (*Sabatini et al., 2021*; *Meek et al., 2013*;

*Meek et al., 2016*). Hence, each VMH subregion may contain multiple discrete neuron populations that mediate unique functions.

To date, most analyses of VMH function have utilized *Nr5a1* or candidate markers that do not necessarily align with functionally and/or transcriptionally unique VMH cell types (*Bingham et al., 2006*). Thus, to understand VMH-controlled responses, we must use unbiased methods to define discrete subpopulations of VMH neurons, along with markers that permit their selective manipulation. Single-cell approaches (such as single-nucleus RNA-sequencing [snRNA-seq]) can identify neuronal populations in an unbiased manner and have previously suggested parceling schemes for neurons in many brain areas, including the VMH (*Kim et al., 2019*; *Campbell et al., 2017*; *Habib et al., 2017*). Many such analyses define large numbers of highly interrelated cell populations

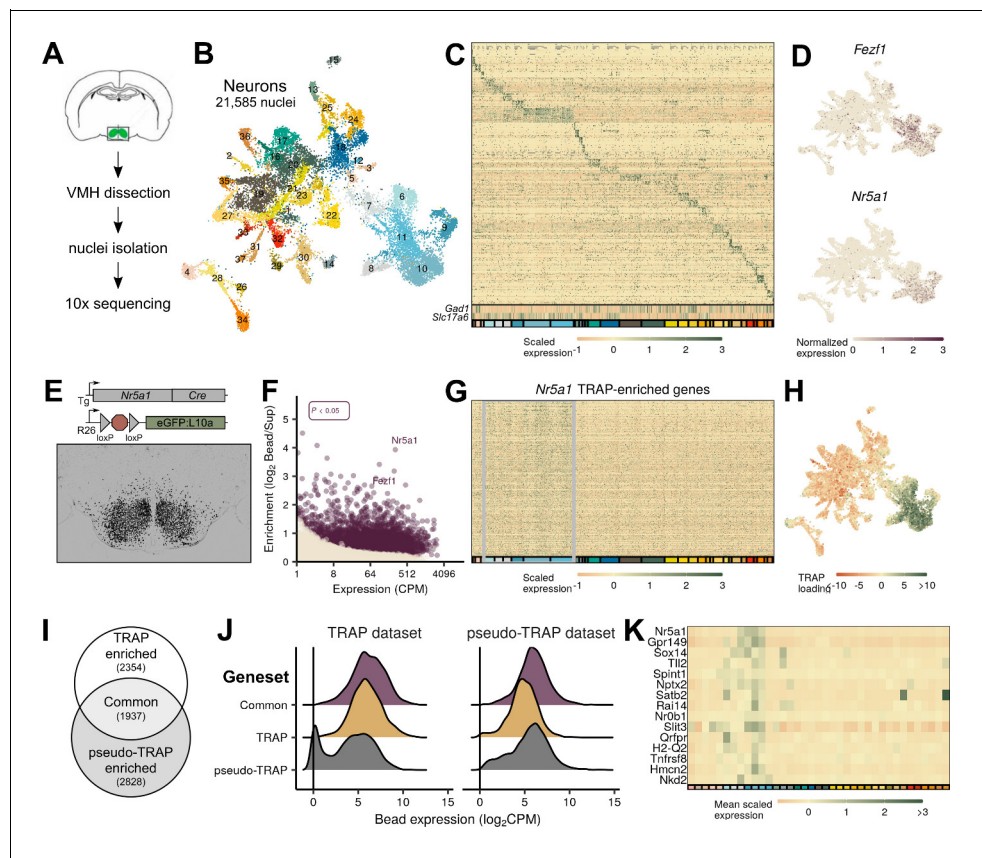

**Figure 1.** Identification of ventromedial hypothalamic nucleus (VMH) neurons from mice. (**A**) Schematic of VMH isolation and single-cell sequencing protocol. (**B**) UMAP of 21,585 neuronal nuclei colored and labeled by cluster designation. (**C**) Expression profile of the top enriched genes for each cluster (colored on bottom), including GABAergic (*Gad1*) and glutamatergic (*Slc17a6*) markers. (**D**) Expression of *Nr5a1* and *Fezf1* in individual cells in UMAP space. (**E**) *Nr5a1-Cre* translating ribosome affinity purification with RNA-sequencing (TRAP-seq) overview. *Nr5a1-Cre* mice were crossed with *ROSA26^eGFP-L10a* mice, resulting in VMH-restricted eGFP-L10a expression. Representative image shows GFP-IR (black) in a coronal section from these mice. (**F**) TRAP-seq revealed the enrichment of thousands of genes (including *Nr5a1* and *Fezf1*) in these cells relative to non-TRAP material. (**G**) Expression profile of the top enriched genes from *Nr5a1-Cre* TRAP-seq across clusters; gray box indicates presumptive VMH cells. (**H**) Magnitude of the first principal component after performing principal components analysis for the genes enriched in *Nr5a1-Cre* TRAP-seq. (**I**) Venn diagram of genes enriched in *Nr5a1-Cre* TRAP-seq (TRAP enriched), in single-nucleus RNA-sequencing (snRNA-seq) VMH pseudo-TRAP (pseudo-TRAP enriched), or both (common). Number in parentheses refers to the number of genes in each category. (**J**) Histograms of expression level for genes by enrichment gene set in each dataset (*Nr5a1-Cre* TRAP-seq or snRNA-seq). (**K**) Mean scaled expression for each cluster for the top genes enriched in *Nr5a1-Cre* TRAP-seq.

The online version of this article includes the following figure supplement(s) for figure 1:

**Figure supplement 1.** Mouse single-nucleus RNA-sequencing (snRNA-seq) identifies major CNS classes.

of unclear functional significance and conservation, however (*Kim et al., 2019*; *Chen et al., 2017*; *Lam et al., 2017*). Determining functions for dozens of cell populations that lie in the same anatomic region and which possess overlapping gene expression profiles (i.e., that do not contain unique marker genes) would represent a daunting task.

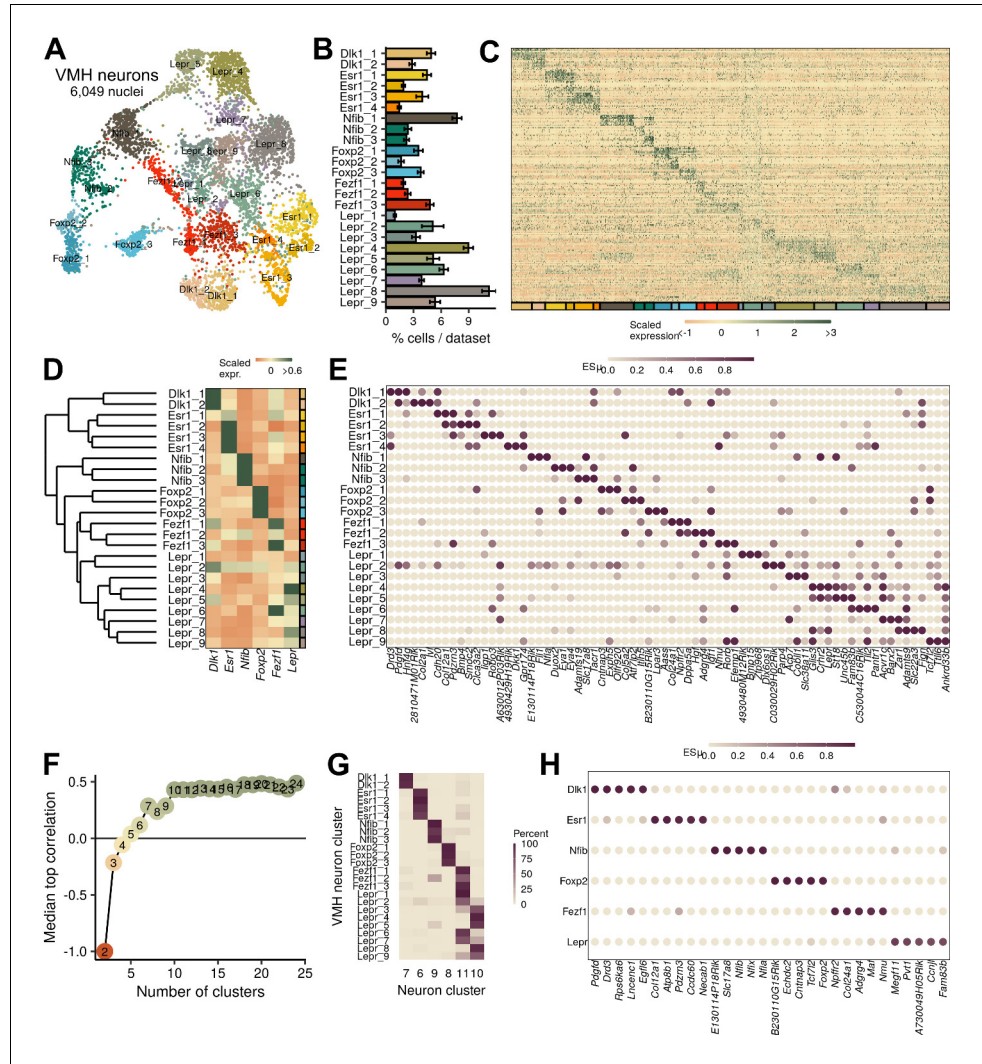

**Figure 2.** Ventromedial hypothalamic nucleus (VMH) neuronal populations can be grouped into six major classes. (**A**) UMAP of 6049 VMH neurons colored and labeled by cluster designation. (**B**) Prevalence of clusters across samples, mean ± SEM. (**C**) Expression profile of the top enriched genes for each cluster. (**D**) Hierarchical clustering and mean expression of marker genes for each class of neurons. (**E**) ESμ for the top three marker genes for each population determined by CELLEX. (**F**) Median maximal pairwise expression correlation for each cut of the hierarchical tree resulting in 2–24 clusters. (**G**) Percent of cells in each VMH cluster that correspond to each neuronal cluster (from *Figure 1*). (**H**) ESμ for the top five marker genes for each major class determined by CELLEX.

The online version of this article includes the following figure supplement(s) for figure 2:

**Figure supplement 1.** Comparison with ventromedial hypothalamic nucleus (VMH) data from Kim et al.

**Figure supplement 2.** Comparison of data with ventromedial hypothalamic nucleus (VMH) data from Campbell et al.

**Figure supplement 3.** Identification of ventromedial hypothalamic nucleus (VMH) neuronal 'classes'.

**Figure supplement 4.** *Dlk1* is expressed in neurons adjacent to the ventromedial hypothalamic nucleus (VMH).

**Figure supplement 5.** Ventromedial hypothalamic nucleus (VMH) *Nfib* population localizes to dorsomedial compartment.

**Figure supplement 6.** Foxp2 population localizes to anterolateral ('tuberal') compartment.

In the present study we use translating ribosome affinity purification with RNA-sequencing (TRAP-seq) in mice together with snRNA-seq of mouse and macaque VMH neurons to define transcriptionally unique, anatomically discrete, conserved, and genetically tractable classes of VMH neurons. These include a distinct *Lepr*-expressing VMH neuron class, along with a set of glutamatergic VMH neurons that resides in the tuberal region. These findings define a starting point for the comprehensive analysis of VMH circuitry and function.

## Results

### Combining snRNA-seq with *Nr5a1*-directed TRAP-seq defines mouse VMH neuron populations

To define neuronal populations within the mouse VMH in an unbiased manner, we microdissected the VMH of mice and subjected 10 individual tissue samples to snRNA-seq using the $10\times$ Genomics platform (*Figure 1A*), collecting a total of 42,040 nuclei that passed quality control (*Figure 1—figure supplement 1A–C*). The recovered nuclei included all major CNS cell types (*Figure 1—figure supplement 1*, see Materials and methods for clustering and cell-type identification details), including 21,585 neurons that comprised 37 distinct neuronal populations (*Figure 1B,C*).

Many adult VMH neurons express *Nr5a1* (which encodes the transcription factor, SF1; *Cheung et al., 2013*) and/or *Fezf1* (*Kurrasch et al., 2007*), whose detection was restricted to a confined cluster of neurons in UMAP space (*Figure 1D*). Although essentially all VMH neurons express *Nr5a1* during development, only a subset of VMH cells express *Nr5a1* and/or *Fezf1* in adult animals (*Cheung et al., 2013*; *Kurrasch et al., 2007*). Furthermore, the inherent noise in snRNA-seq data risks false positives and negatives when using only one or two genes for cell-type identification. To ensure that we identified all VMH cell groups for our analysis, we performed TRAP-seq using *Nr5a1-Cre;Rosa26eGFP-L10a* mice, in which the early developmental expression of *Nr5a1-Cre* promotes the permanent expression of tagged ribosomes across the VMH (*Figure 1E*). TRAP-seq identified 4492 transcripts significantly enriched in cells marked by the developmental expression of *Nr5a1*, including *Nr5a1* and *Fezf1* (*Figure 1F*, *Supplementary file 1*). Applying this broader VMH-enriched transcriptome to our snRNA-seq clusters revealed six populations of neurons (clusters 6–11; corresponding to the populations with highest *Nr5a1* and *Fezf1* expression) that contain VMH neurons (*Figure 1G,H*). Importantly, *Nr5a1*-negative cells (clusters 6 and 7) were identified as VMH by their enrichment of *Nr5a1-Cre* TRAP-seq genes (*Figure 1H*), suggesting they were developmentally labeled by *Nr5a1-Cre*; these cells presumably reside in the VMH$_{VL}$ (which expresses *Nr5a1* during development, but not in adulthood).

To compare TRAP-seq and snRNA-seq results, we performed 'pseudo-TRAP' on pseudobulk samples of our snRNA-seq data, aggregated by cell type (i.e., VMH clusters vs. non-VMH clusters; see *Supplementary file 2* for enrichment results). While many genes were enriched in both datasets (1977), more were specific to one method (*Figures 1I*, 2354 genes specific to TRAP and 2828 specific to pseudo-TRAP). Notably, enrichment was largely a function of expression level: the genes that were enriched in both datasets were highly expressed in both, while the genes that were specific to TRAP or pseudo-TRAP were highly expressed in their enriched dataset, but expressed at lower levels in the other (*Figure 1J*). This suggests that the two methods may illuminate partially distinct aspects of the transcriptome—TRAP-seq for ribosome-associated genes and snRNA-seq for nuclear-enriched and nascent transcripts—and that their combined use provides a more comprehensive view of cellular state. Finally, many of the genes enriched in both datasets were limited to one or a few populations, highlighting the heterogeneity of gene expression across VMH cell types, even for prominent VMH marker genes (*Figure 1K*).

To understand the landscape of mouse VMH neuron populations in more detail, we subjected the VMH neurons in clusters 6–11 (*Figure 1H*) to further clustering, which identified 24 transcriptionally defined neuronal populations (*Figure 2A–C*). The cell groups that we identified are largely consistent with a recently published VMH$_{VL}$-focused single-cell RNA-seq study (*Figure 2—figure supplement 1*; *Kim et al., 2019*). While our study identified 24 clusters whose major markers were evenly distributed between the dorsomedial and ventrolateral compartments of the VMH, Kim et al. identified 31 clusters with a bias toward populations with VMH$_{VL}$ markers. Integrating the datasets (*Figure 2—figure supplement 1*) revealed their broad correspondence, with highly correlated

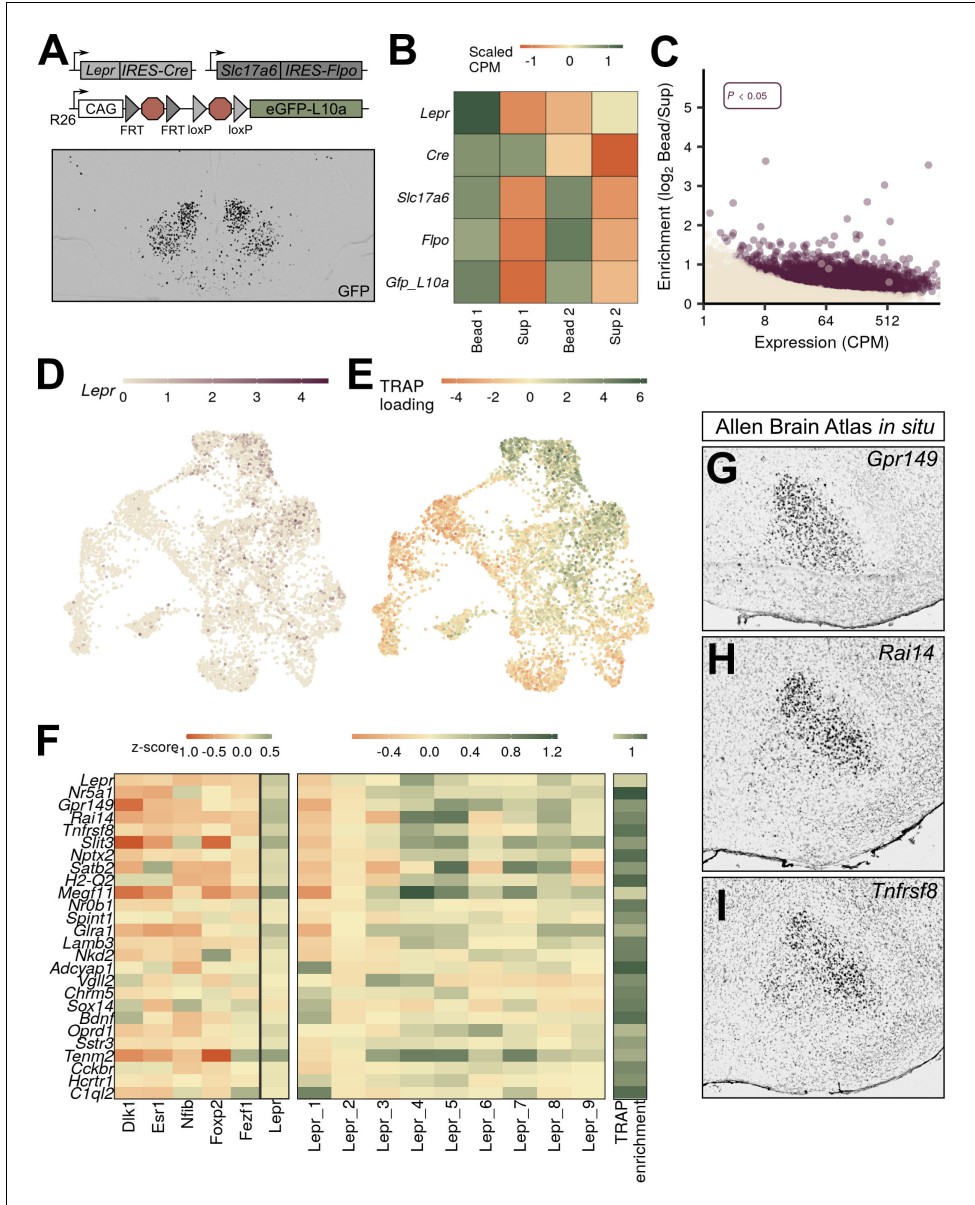

**Figure 3.** VMH<sup>Lepr</sup> neurons represent a distinct class of ventromedial hypothalamic nucleus (VMH) neurons. (**A**) Diagram of strategy to transcriptionally profile the VMH<sup>Lepr</sup> neurons by crossing *Lepr<sup>Cre</sup>* and *Slc17a6<sup>Flpo</sup>* to a mouse line in which the *ROSA26* (R26) locus contains a CAG-driven, Flp- and Cre-dependent *eGFP:L10a* allele (*RCFL<sup>eGFP-L10a</sup>*). (Below) A representative image of GFP-IR (black) expression in *Lepr<sup>Cre</sup>;Slc17a6<sup>Flpo</sup>;RCFL<sup>eGFP-L10a</sup>* (Lepr<sup>Slc17a6</sup>-L10a) mice. (**B**) Scaled counts per million (CPM) for each gene in Lepr<sup>Slc17a6</sup>-L10a mice. (**C**) Expression and enrichment of genes from Lepr<sup>Slc17a6</sup>-L10a VMH pulldown. (**D**) Expression of *Lepr* in individual VMH neurons in UMAP space. (**E**) Magnitude of the first principal component after performing principal components analysis for the genes enriched in Lepr<sup>Slc17a6</sup>-L10a VMH translating ribosome affinity purification with RNA-sequencing (TRAP-seq), projected into UMAP space. (**F**) Mean class expression (left), Lepr cluster expression (center), and Lepr<sup>Slc17a6</sup>-L10a TRAP-seq enrichment (right) of the top genes unique to the VMH<sup>Lepr</sup> population by both TRAP and pseudo-TRAP. (**G–I**) Sagittal Allen Brain Atlas in situ images for (**G**) *Gpr149*, (**H**) *Rai14*, and (**I**) *Tnfrsf8*; all probes shown in black.

The online version of this article includes the following figure supplement(s) for figure 3:

**Figure supplement 1.** Comparison of different translating ribosome affinity purification with RNA-sequencing (TRAP-seq) approaches for identifying genes enriched in *Lepr* ventromedial hypothalamic nucleus (VMH) cells.

expression profiles (*Figure 2—figure supplement 1B*) and shared marker genes (*Figure 2—figure supplement 1D*). The omission of *Nfib*-marked populations from the (*Kim et al., 2019*) analysis represented a notable difference between our analyses, however. In our reanalysis of their data, we found that *Nfib*-marked cells were present in their samples but were filtered out before the final VMH clustering (*Figure 2—figure supplement 1*). A previous scRNA-seq study of the neighboring ARC also mapped a neuron population marked by *Nfib* expression to the VMH, however (*Campbell et al., 2017*), and these cells correlated to our *Nfib*-marked VMH neuron populations (*Figure 2—figure supplement 2*).

## A simplified parceling scheme defines anatomically distinct VMH neuron populations

Hierarchical clustering and marker gene analysis for our 24 mouse VMH neuron clusters using CEL-LEX (*Timshel et al., 2020*) revealed that many cell groups were highly related to other VMH neuron clusters (*Figure 2D,E*; *Supplementary file 3*). Furthermore, many of these populations share marker gene expression to an extent that renders it impossible to specifically manipulate single populations given current approaches that use a single gene for cell-type manipulation (e.g., Cre-based mouse models) (*Figure 2D,E*). To identify classes of genetically distinguishable cells, we cut the hierarchical tree at different levels and measured the maximum pairwise expression correlation to highlight the level at which few pairs of clusters exhibited highly correlated transcriptomes. We found that six classes represented the largest number of classes that retained minimal correlated expression (*Figure 2F*). To avoid confusion, we refer to these as VMH neuron classes, while referring to the cell groups of which each class is composed as clusters or subpopulations. The mean silhouette width (a measure of clustering robustness) for the various tree cuts also supported the use of six classes (*Figure 2—figure supplement 3*), and these six classes corresponded to the cluster designations from the broader neuron dataset (*Figure 2G*). Hence, a parceling scheme for mouse VMH neurons that

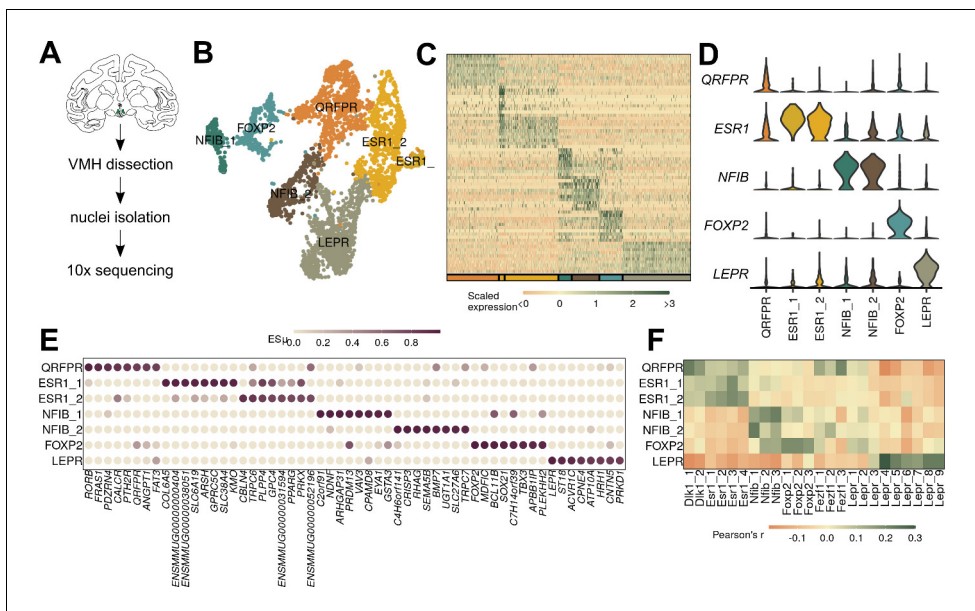

**Figure 4.** Macaque ventromedial hypothalamic nucleus (VMH) populations revealed by single-nucleus RNA-sequencing (snRNA-seq). (A) Schematic of experimental process for macaque snRNA-seq. (B) UMAP of 3752 VMH neuronal nuclei colored and labeled by cluster designation. (C) Expression profile of the top enriched genes for each cluster. (D) Violin plot of normalized expression for marker genes for each VMH neuronal population. (E) ESµ for the top five marker genes for each cluster determined by CELLEX. (F) Pairwise scaled expression correlation (Pearson's r) for each macaque and mouse VMH neuronal cluster.

The online version of this article includes the following figure supplement(s) for figure 4:

**Figure supplement 1.** Macaque single-nucleus RNA-sequencing (snRNA-seq) identifies major CNS classes.
**Figure supplement 2.** Identifying ventromedial hypothalamic nucleus (VMH) neurons in macaque.

contains six classes, each composed of highly similar subpopulations, captures the transcriptional patterns of the VMH. Importantly, this approach identified numerous specific marker genes (e.g., *Dlk1*, *Esr1*, *Nfib*, *Foxp2*, *Fezf1*, and *Lepr*) for each class of mouse VMH neurons (*Figure 2D,H*, *Supplementary file 4*), which should facilitate their manipulation and study.

Consistent with the distinct nature of these six VMH neuron classes and the utility of this parceling scheme, each class demonstrated a circumscribed anatomic distribution. As previously reported (*Lee et al., 2014*; *Persson-Augner et al., 2014*), *Dlk1*- and *Esr1*-expressing neurons (VMH^Dlk1 and VMH^Esr1, respectively) map to the VMH_VL. While *Dlk1* is unique to a single class of VMH neurons, it is also expressed in neighboring hypothalamic neurons (*Figure 2—figure supplement 4*), complicating its utility for manipulating this population without using an intersectional approach. In contrast to

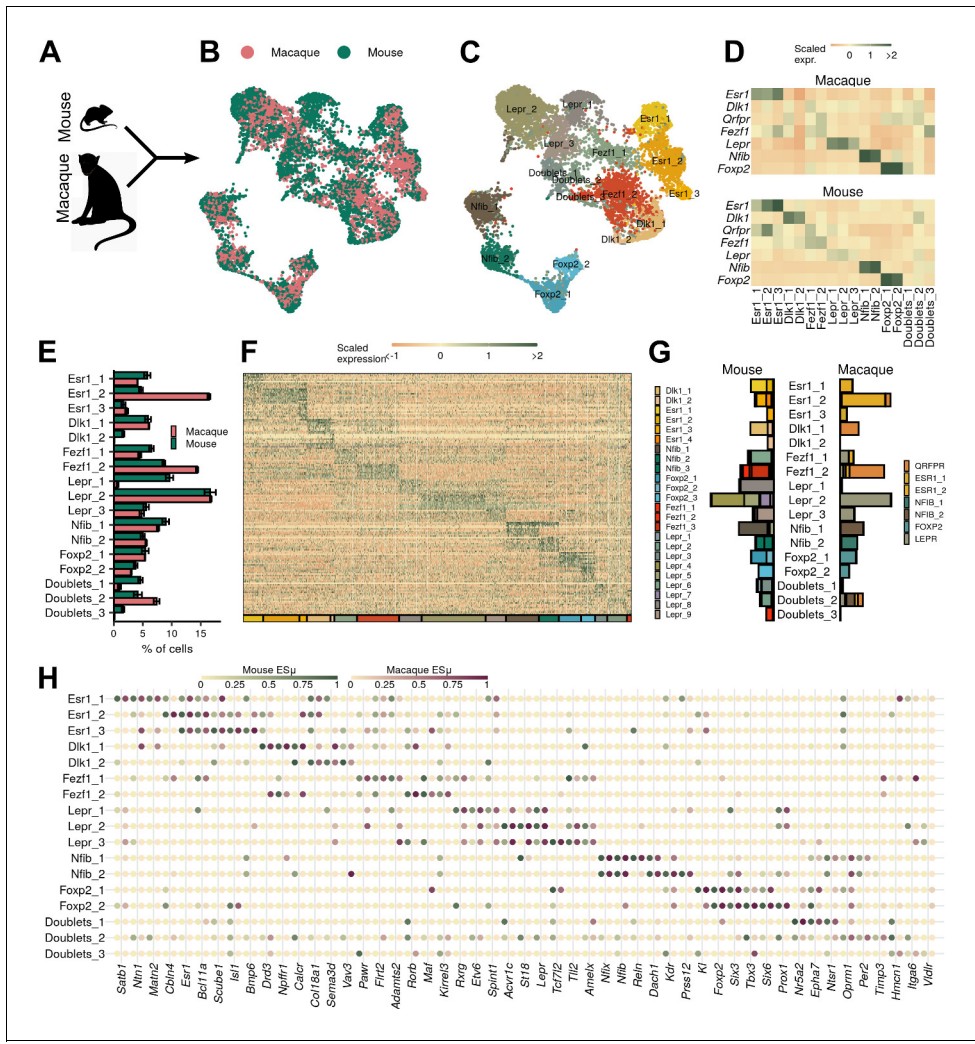

**Figure 5.** Ventromedial hypothalamic nucleus (VMH) populations are conserved between mouse and macaque. (A–B) Mouse and macaque single-nucleus RNA-sequencing (snRNA-seq) datasets were (A) merged using canonical correlation analysis and (B) projected onto UMAP space, colored here by species. (C) UMAP of VMH neuronal nuclei colored and labeled by cluster designation. (D) Mean scaled expression of marker genes across integrated clusters by species. (E) Proportion of cells in each cluster from the sample for each species (mean ± SEM). (F) Expression profile of the top enriched genes for each cluster. (G) Mapping of species-specific clusters onto the integrated clusters. (H) Species-specific ESμ for the top three marker genes for each integrated cluster determined by CELLEX.

The online version of this article includes the following figure supplement(s) for figure 5:

**Figure supplement 1.** Similarities of mouse and macaque clusters.

**Figure supplement 2.** Cluster marker expression in macaque ventromedial hypothalamic nucleus (VMH).

these ventrolateral populations, *Lepr*-expressing cells lie within the core of the VMH$_{DM}$ (*Elmquist et al., 1998*) and the expression of marker genes for the *Nfib*-marked clusters (VMH$^{Nfib}$) resides in the most dorsomedial compartment of the VMH$_{DM}$ (*Figure 2—figure supplement 5*). *Fezf1*-marked populations (VMH$^{Fezf1}$) include cells with similarity to both ventrolateral VMH$^{Dlk1}$ and dorsomedial VMH$^{Lepr}$ neurons, and many VMH$^{Fezf1}$ neurons lie in VMH$_C$, in the transition between the dorsomedial and ventrolateral zones of the VMH. Markers for the *Foxp2*-expressing populations (VMH$^{Foxp2}$) reside anterior and lateral to the core of the VMH, in the so-called tuberal region (*Figure 2—figure supplement 6*). Thus, the major VMH classes identified by our analysis each map to specific and distinct anatomic locations, consistent with their unique genetic signatures.

## *Lepr*-directed TRAP-seq analysis of VMH$^{Lepr}$ cells

Given that *Lepr*-expressing VMH neurons mediate only a subset of VMH$_{DM}$ functions, we were intrigued by the finding that unbiased snRNA-seq identified a distinct class of VMH$_{DM}$ cells marked by *Lepr* (VMH$^{Lepr}$ neurons), consistent with a specialized role for *Lepr*-expressing VMH cells (*Minokoshi et al., 1999*; *Toda et al., 2013*; *Noble et al., 2014*; *Gavini et al., 2016*). The finding that *Lepr*-expressing VMH neurons map onto a single VMH neuron class also suggests a uniformity of function for these cells, which contrasts with the situation in other brain regions (such as the neighboring ARC, where multiple cell types with opposing functions [e.g., *Agrp* and *Pomc* cells] express *Lepr*; *Campbell et al., 2017*). We utilized TRAP-seq to assess the extent to which *Lepr*-expressing VMH neurons correspond to the VMH$^{Lepr}$ clusters and to compare the genetic program of *Lepr*-expressing VMH neurons with gene expression in snRNA-seq-defined VMH neuron classes and subpopulations.

TRAP-seq analysis of microdissected VMH tissue from *Lepr$^{Cre}$;Rosa26$^{eGFP-L10a}$* animals (*Leshan et al., 2006*) resulted in the enrichment of transcripts from *Lepr$^{Cre}$* neurons that lie in VMH-adjacent brain areas, including *Agrp*-, *Ghrh*-, *Pomc*-, and *Nts*-expressing cells from the ARC and lateral hypothalamic area (LHA) (*Figure 3—figure supplement 1*). To more closely restrict our TRAP-seq analysis to *Lepr*-expressing neurons that reside in the VMH, we used a recently developed mouse line that expresses eGFP-L10a only in cells that contain both Cre and Flp recombinases (*Sabatini et al., 2021*). Because VMH neurons contain vGLUT2 (encoded by *Slc17a6*), while most *Lepr*-expressing neurons in the neighboring ARC, dorsomedial hypothalamus (DMH), and LHA do not (*Vong et al., 2011*), we crossed an *Slc17a6$^{Flpo}$* mouse line to *Lepr$^{Cre}$* and *RCFL$^{eGFP-L10}$* to produce *Slc17a6$^{Flpo}$; Lepr$^{Cre}$; RCFL$^{eGFP-L10}$* (Lepr$^{Slc17a6}$-L10a) mice. In these mice, mediobasal hypothalamic eGFP-L10a was largely restricted to the VMH (*Figure 3A*; *Sabatini et al., 2021*).

We microdissected the VMH of Lepr$^{Slc17a6}$-L10a mice and performed TRAP-seq, identifying 3580 transcripts that were enriched in the TRAP material relative to the supernatant (*Figure 3C*, *Supplementary file 5*). Importantly, *Lepr* itself was not enriched, suggesting that we successfully purified ribosome-associated mRNA from *Lepr*-expressing VMH cells away from that derived from other *Lepr*-expressing populations (*Figure 3B*). We found that most non-VMH genes enriched in our conventional (*Lepr$^{Cre}$*-only) TRAP-Seq (including *Agrp*, *Nts*, and *Ghrh*) were not enriched in this analysis (*Figure 3—figure supplement 1*). *Pomc* and *Prlh* remained somewhat enriched, however, suggesting that *Lepr*-expressing DMH *Prlh* cells are glutamatergic (*Dodd et al., 2014*), and consistent with the finding that some ARC *Pomc* cells express *Slc17a6* (*Jones et al., 2019*).

As expected, we observed a high degree of concordance between snRNA-seq-defined gene expression in neurons of the VMH$^{Lepr}$ cluster with gene expression in *Lepr*-expressing VMH neurons by TRAP-seq. We identified 3576 genes enriched in VMH-centered Lepr$^{Slc17a6}$ TRAP-seq material (*Figure 3C*), 1174 of which were also enriched in pseudo-TRAP analysis of the neurons assigned to the VMH$^{Lepr}$ population (*Supplementary file 6*). Among the top enriched genes shared by these two methods were *Gpr149*, *Rai14*, and *Tnfrsf8* (*Figure 3F*), which exhibit a similar VMH$_{DM}$-centered expression pattern as *Lepr* (*Figure 3G–I*). As with the *Nr5a1-Cre* TRAP-seq, enrichment was largely a function of expression level: the genes that were enriched in both datasets were highly expressed in both, while the genes that were specific to TRAP or pseudo-TRAP were highly expressed in their enriched dataset, but more lowly expressed in the other. Notably, gene ontology (GO) analysis of the common genes revealed many terms related to synaptic function, while genes unique to TRAP-seq were enriched for ribosomal and mitochondrial function (data not shown).

We mapped enriched genes from Lepr$^{Slc17a6}$ TRAP-seq to the gene expression profiles of our snRNA-seq-defined VMH neuron populations (*Figure 3F*), revealing the bias of Lepr$^{Slc17a6}$ TRAP-seq

gene expression toward VMH^Lepr neurons and the exclusion of markers from other VMH^Lepr cells from Lepr^Slc17a6 TRAP-seq-enriched genes. Thus, this analysis demonstrates that *Lepr*-expressing VMH neurons map specifically to VMH^Lepr cell clusters, suggesting that they represent a transcriptionally and functionally unique set of neurons.

## Conservation of VMH neuronal populations across species

While many previous studies of VMH neuron function have suggested that this brain region contains neurons that could represent therapeutic targets to aid people with obesity, diabetes, and other diseases, most of these studies have been performed in mice (*Hashikawa et al., 2017*; *Flak et al., 2020*; *Meek et al., 2013*; *Kim et al., 2012*). We know little about the cross-species conservation of VMH cell populations, however. To assess the potential conservation of VMH neuron populations across species, we microdissected macaque (*Macaca mulatta*) VMH and performed snRNA-seq using the same techniques as for mouse VMH (*Figure 4—figure supplement 1*). A subset of macaque neurons expressed *NR5A1* and/or *FEZF1* and exhibited similar gene expression profiles to our mouse *Nr5a1-Cre* TRAP-seq enriched genes (*Figure 4—figure supplement 2*), suggesting that these cells represent the macaque VMH and indicating that the mouse and macaque VMH share similar global gene expression signatures.

Graph-based clustering of the macaque VMH neurons yielded seven populations with unique marker genes (with the partial exception of two related *ESR1*-expressing cell types) (*Figure 4B–E*). Most of these macaque populations, including populations marked by *LEPR*, *FOXP2*, *NFIB,* and *ESR1* (VMH^LEPR, VMH^FOXP2, VMH^NFIB, and VMH^ESR1, respectively), have presumptive orthologs in the mouse (*Figure 4D*, see *Supplementary file 7* for a complete list of markers). While populations marked by *DLK1* and *FEZF1* were absent from this analysis, the macaque VMH contained a population marked by *QRFPR* expression (VMH^QRFPR).

To determine whether the macaque populations were orthologous to those from the mouse, we first compared expression of orthologous genes among mouse and macaque cell clusters. We found that all macaque populations had clear correlates in the mouse, which mapped according to their marker genes (as expected) (*Figure 4F*, *Figure 5—figure supplement 1D*). Notably, the macaque VMH^QRFPR population correlated with clusters from both VMH^Dlk1 and VMH^Fezf1 classes in the mouse, suggesting that VMH^QRFPR contains orthologs of the mouse VMH^Dlk1 and VMH^Fezf1 classes. Projecting the mouse or macaque cluster labels onto the other species using Seurat anchors (see Materials and methods for more details) yielded similar results (*Figure 5—figure supplement 1*). All major classes from mouse and macaque projected with high confidence onto the equivalent major classes of the other species, and the macaque VMH^QRFPR population was represented by both mouse VMH^Dlk1 and VMH^Fezf1 cells (*Figure 5—figure supplement 1A*).

To generate an atlas of conserved mouse and macaque VMH populations, we integrated the mouse and macaque data and clustered the merged dataset using the Seurat canonical correlation analysis (CCA) framework (*Figure 5A–B*, see Materials and methods for more details). This analysis revealed populations of VMH neurons that each contained mouse and macaque cells in roughly equal proportions (*Figure 5E*). As predicted, neurons from the macaque VMH^LEPR, VMH^FOXP2, VMH^NFIB, and VMH^ESR1 classes mapped directly with murine VMH^Lepr, VMH^Foxp2, VMH^Nfib, and VMH^Esr1 classes, respectively (*Figure 5G*). We examined the potential co-expression of conserved marker genes for VMH^LEPR cells by in situ hybridization (ISH) for *ACVR1C* in the macaque hypothalamus (*Figure 5—figure supplement 2*), confirming *ACVR1C* is co-expressed with *LEPR* in the macaque VMH_DM. We also examined the potential co-expression of *SLC17A8* (which marks VMH^NFIB) with *LEPR*. *SLC17A8* identified a population of cells that lay at the medial extreme of the macaque VMH_DM, corresponding to the most dorsomedial aspect of the rodent VMH_DM, as for murine VMH^Nfib cells; these cells did not colocalize with *LEPR*-expressing cells (*Figure 5—figure supplement 2*).

This analysis also identified macaque populations that mapped with the murine VMH^Dlk1 and VMH^Fezf1 populations; these derived mainly from the macaque VMH^QRFPR population (*Figure 5G*). Notably, *FEZF1* and *DLK1* were poorly enriched in the macaque VMH^QRFPR population (*Figure 5D*), while mouse *Qrfpr* expression was biased toward the VMH^Esr1 populations (*Figure 5D*). Despite differences in some marker genes, however, the orthologous macaque and mouse VMH neuron populations share dozens of other genes across species (*Figure 5H*, *Supplementary file 8*, *Figure 5—figure supplement 1B*), suggesting that the mouse and macaque cells in each group represent

similar cell types. Gene ontology (GO) analysis of common marker genes (*Source data 2*, *Source data 1*) revealed that most of these mediate core neuronal functions, such as ion channel activity (*Figure 5—figure supplement 1C*). Thus, while some marker genes vary across species, the mouse VMH populations have close orthologs in the macaque VMH based upon their gene expression profiles.

## Discussion

We combined TRAP- and snRNA-seq analysis of the VMH to identify 24 mouse neuronal populations with complex interrelations. These 24 populations represented six distinct cell classes that demonstrated unique anatomic distribution patterns. The main VMH classes were highly conserved between the mouse and the macaque in terms of gene expression profiles and anatomic distribution within the VMH. This atlas of conserved VMH neuron populations provides an unbiased and tractable starting point for the analysis of VMH circuitry and function.

Having many populations of VMH neurons with highly related gene expression profiles complicates the functional analysis of VMH cell types suggesting the importance of simplifying the map of these heterogeneous populations to permit their manipulation and study. By using hierarchical clustering, we were able to identify six maximally unrelated classes of VMH neurons with distinct gene expression signatures. These six discrete transcriptionally defined VMH neuronal classes demonstrated distinct anatomic distribution patterns (three located in the $VMH_{DM}$, two in the $VMH_{VL}$, and one in the tuberal region), revealing a transcriptional basis for the previously suggested functional architecture of the VMH.

Our VMH cell populations were similar to those previously described by Kim et al., although they more finely split the $VMH_{VL}$ populations than did we (presumably because their dissection bias toward the $VMH_{VL}$ yielded more $VMH_{VL}$ neurons) (*Kim et al., 2019*). We were able to more clearly distinguish among $VMH_{DM}$ cell types in our $VMH_{DM}$-focused analysis, however, including by dividing the $VMH_{DM}$ neurons that they identified into two major classes ($VMH^{Lepr}$ and $VMH^{Fezf1}$), as well as identifying a third $VMH_{DM}$ population ($VMH^{Nfib}$) absent from their analysis. Interestingly, the $VMH^{Nfib}$ population was quite distinct from the more closely related $VMH^{Lepr}$ and $VMH^{Fezf1}$ $VMH_{DM}$ cell types, suggesting a potentially divergent function for this most dorsomedial of the $VMH_{DM}$ populations.

Our mouse VMH cell types mapped clearly onto specific populations of macaque VMH neurons, revealing the utility of the mouse as a surrogate for the primate in terms of VMH cell types and, presumably, function. While the macaque single $VMH^{QRFRP}$ population mapped to two mouse classes ($VMH^{Fezf1}$ and $VMH^{Dlk1}$) by orthologous gene expression, these mouse classes are closely related to each other (transcriptionally and anatomically) and the macaque cells from $VMH^{QRFRP}$ segregate to distinct cell clusters defined by *Dlk1* and *Fezf1* in the mouse when we integrated and reclustered the mouse and macaque cell data. Also, while *DLK1* is not specific to the macaque $VMH^{QRFPR}$ cells, the otherwise similar gene expression profiles of mouse and macaque cells that map to $VMH^{Dlk1}$ or $VMH^{Fezf1}$ populations suggest the conserved nature of these cell types across species.

While not all *Lepr*-expressing cell types track in the brain across species (e.g., preproglucagon [*Gcg*]-containing NTS neurons in the mouse express *Lepr*, rat NTS *Gcg* cells do not; *Huo et al., 2008*), mouse $VMH^{Lepr}$ neurons map directly with macaque $VMH^{LEPR}$ neurons by all of the measures that we examined. The finding that *Lepr/LEPR* expression marks a unique and conserved cell type in rodent and primate is consistent with the notion that this class of VMH neuron mediates a discrete component of VMH function, as suggested by previous work in the mouse demonstrating roles for *Lepr*-expressing VMH neurons in the control of energy balance, but not glucose production or panic-like behaviors (*Meek et al., 2013*; *Meek et al., 2016*; *Sabatini et al., 2021*).

While the tuberal region contains more GABAergic than glutamatergic neurons, this region projects to similar target areas as does the core VMH and contains substantial numbers of neurons marked by *Nr5a1-Cre* activity. Hence, although $VMH^{Foxp2}$ cells lie in the tuberal region, their glutamatergic nature, their marking by *Nr5a1-Cre* activity, and the finding that they are transcriptional most similar to other VMH populations mark $VMH^{Foxp2}$ cells as VMH neurons. While few data exist to suggest the physiologic roles played by these cells, it will be interesting to manipulate $VMH^{Foxp2}$ neurons to determine their function.

The identification of distinct transcriptionally defined VMH cell populations provides the opportunity to develop new tools that can be used to understand the nature and function of VMH$_{DM}$ cell types and their roles in metabolic control. The finding that the major VMH cell classes found in the mouse are present in the macaque supports the use of the mouse to study the metabolic functions of the VMH as a means to identify potential therapeutic targets for human disease. It will be important to use these findings to dissect functions for subtypes of VMH cells, which may represent targets for the therapy of diseases including obesity and diabetes.

## Materials and methods

### Animals

Mice were bred in the Unit for Laboratory Animal Medicine at the University of Michigan. These mice and the procedures performed were approved by the University of Michigan Committee on the Use and Care of Animals and in accordance with Association for the Assessment and Approval of Laboratory Animal Care (AAALAC) and National Institutes of Health (NIH) guidelines. Mice were provided with ad libitum access to food (Purina Lab Diet 5001) and water in temperature-controlled (25°C) rooms on a 12 hr light-dark cycle with daily health status checks.

*Nr5a1-Cre* (Jax: 012462) (*Dhillon et al., 2006*) and *Foxp2$^{IRES-Cre}$* (Jax: 030541) (*Rousso et al., 2016*) mice were obtained from Jackson Laboratories. *Rosa26* $^{CAG-LSL-eGFP-L10a}$, *Lepr$^{Cre}$* (Jax: 032457), *Slc17a6$^{Flpo}$*, and *RCFL$^{eGFP-L10a}$* mice have been described previously (*Leshan et al., 2006*; *Krashes et al., 2014*; *Sabatini et al., 2021*).

Nonhuman primate tissue was obtained from the Tissue Distribution Program at ONPRC. Animal care is in accordance with the recommendations described in the Guide for the Care and Use of Laboratory Animals of the NIH and animal facilities at the Oregon National Primate Research Center (ONPRC) are accredited by AAALAC.

### Tissue prep, cDNA amplification, and library construction for 10× snRNA-seq

Mice were euthanized using isoflurane and decapitated, the brain was subsequently removed from the skull and sectioned into 1-mm-thick coronal slices using a brain matrix. The VMH was dissected out and flash frozen in liquid N$_2$. Nuclei were isolated as previously described (*Habib et al., 2017*) with modifications as follows. On the day of the experiment, frozen VMH (from 2 to 3 mice) was homogenized in Lysis Buffer (EZ Prep Nuclei Kit, Sigma) with Protector RNAase Inhibitor (Sigma) and filtered through a 30 µm MACS strainer (Myltenti). Strained samples were centrifuged at 500 rcf × 5 min and pelleted nuclei were resuspended in wash buffer (10 mM Tris Buffer pH 8.0, 5 mM KCl, 12.5 mM MgCl$_2$, 1% BSA with RNAse inhibitor). Nuclei were strained again and recentrifuged at 500 rcf × 5 min. Washed nuclei were resuspended in wash buffer with propidium iodide (Sigma) and stained nuclei underwent FACS sorting on a MoFlo Astrios Cell Sorter. Sorted nuclei were centrifuged at 100 rcf × 6 min and resuspended in wash buffer to obtain a concentration of 750–1200 nuclei/µL. RT mix was added to target ~10,000 nuclei recovered and loaded onto the 10× Chromium Controller chip. The Chromium Single Cell 3′ Library and Gel Bead Kit v3, Chromium Chip B Single Cell kit, and Chromium i7 Multiplex Kit were used for subsequent RT, cDNA amplification, and library preparation as instructed by the manufacturer. Libraries were sequenced on an Illumina HiSeq 4000 or NovaSeq 6000 (pair-ended with read lengths of 150 nt) to a depth of at least 50,000 reads/cell.

### snRNA-seq data analysis

Count tables were generated from the FASTQ files using cellranger and analyzed in R 3.6.3 using the Seurat three framework. Genes expressed in at least four cells in each sample and were not gene models (starting with 'Gm') or located on the mitochondrial genome were retained. Cells with at least 500 detected genes were retained. Doublets were detected using Scrublet (*Wolock et al., 2019*). For each 10× run, the expected number of doublets was predicted using a linear model given 10× data of the detected doublet rate and the number of cells. Then, each cell was given a doublet score with Scrublet and the n cells (corresponding to the expected number of doublets) with the top scores were removed.

The data was then normalized using scran (*Lun et al., 2016*) and centered and scaled for each dataset independently and genes that were called variable by both Seurat *FindVariableFeatures* and sctransform (*Hafemeister and Satija, 2019*) were input to principal component analysis (PCA). The top principal components (PCs) were retained at the 'elbow' of the scree plot (normally 15–30, depending on the dataset) and then used for dimension reduction using UMAP and clustering using the Seurat *FindNeighbors* and *FindClusters* functions. Both were optimized for maximizing cluster consistency by clustering over a variety of conditions: first, varying the number of neighbors from 10 to the square root of the number of cells while holding the resolution parameter in *FindClusters* at one and finding the clustering that maximized the mean silhouette score; then, this number of neighbors was held fixed while varying the resolution parameter in *FindClusters* from 0.2 upward in steps of 0.2 until a maximal mean silhouette score was found. Clusters were then hierarchically ordered based on their Euclidean distance in PC space and ordered based on their position in the tree.

Marker genes were found using the Seurat function *FindAllMarkers* for each sample with resulting p-values combined using the *logitp* function from the metap package or using CELLEX 1.0.0. Cluster names were chosen based on genes found in this unbiased marker gene search and known marker genes.

From the all-cell data, cell types were predicted using gene set enrichment analysis from the marker genes and a manually curated set of genes known to mark specific CNS cell types. From this, clusters that were highly enriched for markers from two (or more) distinct cell types were labeled as 'doublets' and those with no enrichment were labeled as 'junk', the remaining clusters were labeled based on their lone CNS cell type.

To predict VMH neurons from all neurons, we first found genes significantly enriched in the bead fraction in *Nr5a1-Cre* TRAP-seq (see below for details) and expressed above one count per million. The scaled count matrix containing these genes was then used as input to PCA. The magnitude of the first PC (loading) was then used to generate a VMH similarity score and the clusters that had a high *Nr5a1-Cre* TRAP loading were glutamatergic (express *Slc17a6* and not *Gad1* or *Slc32a1* above the mean value), and expressed either *Nr5a1* or *Fezf1* above the mean value were included as presumptive VMH.

## TRAP-seq analysis

Mice (*Lepr^Cre^;ROSA26^EGFP-L10a^* or *Lepr^Cre^;Slc17a6^FlpO^;ROSA26^EGFP-L10a^*) were euthanized and decapitated, the brain was subsequently removed from the skull and sectioned into 1-mm-thick coronal slices using a brain matrix. The VMH or hypothalamus was dissected and homogenized in lysis buffer. Between 15 and 20 mice were used for TRAP experiments. GFP-tagged ribosomes were immunoprecipitated and RNA isolated as previously described (*Allison et al., 2018*). RNA was subject to ribodepletion and the resultant mRNA was fragmented and copied into first strand cDNA. The products were purified and enriched by PCR to create the final cDNA library. Samples were sequenced on a 50-cycle single end run on a HiSeq 2500 (Illumina) according to manufacturer's protocols.

FASTQ files were filtered using fastq_quality_filter from fastx_toolkit to remove reads with a phred score <20. Then reads were mapped using STAR with a custom genome containing the Ensembl reference and sequences and annotation for *Cre* and *EGFP:L10a* (and *Flpo* in the RCFL::eGFP-L10a dataset). Count tables were generated using the STAR `-quantMode` GeneCounts flag.

Count tables were analyzed in R 3.6.3 and were subject to quality control to ensure read adequate library size (20–30 million reads), enrichment of positive control genes (e.g., *EGFP:L10a* and/or *Cre, Nr5a1*), and appropriate sample similarity in both hierarchical clustering of Euclidean distance and TSNE/UMAP space (e.g., bead samples are more similar to one another than to any sup sample). All samples passed quality control. Enriched genes were determined using DESeq2 including an effect of sample pair in the model to account for pairing of the bead–sup samples (~pair + cells).

## Integration with published data

Count tables from *Kim et al., 2019* were downloaded from https://data.mendeley.com/datasets/ypx3sw2f7c/3 and count tables from *Campbell et al., 2017* were downloaded from https://www.ncbi.nlm.nih.gov/geo/query/acc.cgi?acc=GSE93374. Note: Only the 10× data from Kim et al. was used. The data was then preprocessed and clustered in the same way as above, though some samples were removed from the Kim et al. dataset for low mean read depth that confounded clustering

(Samples 2018_0802, 2018_0803, 2018_0812_1, 2018_0812_2, and 2018_0812_3). For the Kim et al. dataset, cells were clustered as above (*FindNeighbors* then *FindClusters*), and neuron clusters were predicted using WGCNA to identify correlated gene expression modules. The modules contained dozens of genes that mapped clearly onto a small set of clusters in UMAP space and—based on known marker genes—corresponded to the most prevalent cell types in the brain (e.g., neurons, astrocytes, microglia, oligodendrocytes, etc.). For the Campbell et al. data, neurons were labeled in the metadata from the authors, so neuronal barcodes were simply selected based on their annotation. For both neuronal datasets, VMH neurons were predicted using the same procedure as above: clusters that were glutamatergic, *Fezf1/Nr5a1*-expressing, and similar to *Nr5a1-Cre* TRAP-seq. This corroborated the clusters called VMH in both datasets by the original authors, with the exception of the *Nfib* populations in the Kim et al. dataset that was not called VMH and therefore not assigned a cluster name; we refer to these cells as (Missing) in our integrated dataset.

To find shared populations across datasets, we took two approaches. First, variable genes were found for both datasets using the Seurat *FindVariableFeatures* function. Then, the pairwise Pearson's correlation was found for the mean scaled expression in each cluster in each dataset for the set of genes called variable in both datasets. Additionally, we used the Seurat *FindTransferAnchors* and *IntegrateData* functions to generate a merged dataset that was then subjected to PCA, UMAP reduction, and clustering in the same way as above. These new clusters containing cells from both datasets were then manually named using marker genes from the original datasets (e.g., *Dlk1*, *Esr1*, *Satb2*, *Lepr*, *Nfib*, *Foxp2*, etc.).

### Immunostaining

Animals were perfused with phosphate buffered saline (PBS) for 5 min followed by an additional 5 min of 10% formalin. Brains were then removed and post-fixed in 10% formalin for 24 hr at room temperature, before being moved to 30% sucrose for 24 hr for a minimum of 24 hr at room temperature. Brains were then sectioned as 30-μm-thick free-floating sections and stained. Sections were treated with blocking solution (PBS with 0.1% triton, 3% normal donkey serum; Fisher Scientific) for at least 1 hr. The sections were incubated overnight at room temperature in blocking solution containing primary antibodies. The following day, sections were washed and incubated with fluorescent secondary antibodies with species-specific Alexa Fluor-488 or -568 (Invitrogen, 1:250) to visualize proteins. Primary antibodies used include: GFP (1:1000, #1020, Aves Laboratories) and NFIA (1:500, #PA5-35936, Invitrogen). Images were collected on an Olympus BX51 microscope. Images were background-subtracted and enhanced by shrinking the range of brightness and contrast in ImageJ.

### Macaque snRNA-seq

Whole Rhesus macaque brains were obtained from the Tissue Distribution Program at ONPRC. Brains were centered within a chilled brain matrix (ASI Instruments, catalog # MBM-2000C) and 2 mm slices were obtained from rostral to caudal. Slices containing the hypothalamus were placed in saline and the PVH, ARC, VMH, and DMH were punched out and samples were placed in pre-chilled tubes on dry ice. Samples were stored at −80°C until shipment on dry ice. Nuclei were isolated from frozen macaque tissue as described above for mice.

The FASTQ files were mapped to the macaque genome (Mmul_10) using cellranger and count matrix files were analyzed in R using Seurat three as above, with the exception that gene models and genes mapping to the macaque mitochondrial genome were not removed. Macaque neurons were predicted using orthologs of mouse cell-type marker genes and macaque VMH neurons were identified using macaque orthologs of *Nr5a1-Cre* TRAP-seq enriched genes. Orthologs were identified using Ensembl and only 1:1 orthologs were retained.

### Species integration

The mouse and macaque datasets were integrated in a similar way to integrating the Kim et al. and Campbell et al. mouse datasets. First, the macaque genes were renamed to their mouse orthologs and only 1:1 orthologs and genes expressed in both species were retained. Importantly, because our dataset was biased toward the dorsomedial VMH and the Kim et al. dataset was biased toward the ventrolateral VMH, we also included four randomly chosen samples from the Kim et al. data to get a more representative picture of shared VMH populations across species. The data was then

preprocessed, normalized, and scaled in the same way as previously. The mouse and macaque data was then integrated using the Seurat *FindTransferAnchors* and *IntegrateData* functions and marker genes were found that were common across species by running *FindAllMarkers* for each species separately and then using the metap *logitp* function to find genes that are significantly enriched.

## Macaque ISH

Whole Rhesus macaque brains were obtained from the Tissue Distribution Program at ONPRC. Hypothalamic blocks fixed with 4% paraformaldehyde were incubated in glycerol prior to freezing with isopentane. Tissue was sectioned at 25 μm using a freezing stage sliding microtome and freefloating sections were stored in glycerol cryoprotectant at −20°C. Tissue was mounted on slides prior to ISH, which was performed using ACD Bio RNAScope reagents (Multiplex Fluorescent Detection Kit v2, 323100) for Acvr1c (ACD 591481), Slc17a8 (ACD 543821-c2), and Lepr (ACD 406371-C3). Negative and positive control probes were included in all runs. Slides were imaged on an Olympus VS110 Slidescanner and processed using Visiopharm software.

## Statistical analysis

All data is displayed as mean ± SEM. All plotting and statistical analysis was performed using R 3.6.3. Specific statistical tests are listed in the figure legends.

## Resource availability

All mouse strains will be made available upon reasonable request.

## Code availability

All analysis code will be available at https://github.com/alanrupp/affinati-elife-2021 (*Affinati et al., 2021*, copy archived at swh:1:rev:d17662cffc0bcf7d20acd74c553f0e0e7f01654b).

## Acknowledgements

We thank Randy Seeley, Lotte Bjerre Knudsen, Kevin Grove, Mads Tang-Christensen, Christine Bjørn Jensen, and members of the Myers and Olson labs for helpful discussions. Research support was provided by the Michigan Diabetes Research Center (NIH P30 DK020572, including the Molecular Genetics, Microscopy, and Animal Studies Cores), the Marilyn H Vincent Foundation (to MGM), Novo Nordisk A/S (to MGM), ADA 1–19-PDF-099 (to PVS), and NIH DK122660 (to AHA). NHP data was supported by National Institutes of Health Grant P51 OD-11092 for operation of the Oregon National Primate Research Center and DK123115 (PK).

## Additional information

### Competing interests

Chien Li: is an employee of Novo Nordisk A/S. The other authors declare that no competing interests exist.

### Funding

| Funder | Grant reference number | Author |
| --- | --- | --- |
| National Institutes of Health | dk056731 | Martin G Myers |
| Novo Nordisk | | Martin G Myers |
| National Institutes of Health | P30 DK020572 | Martin G Myers |
| National Institutes of Health | DK122660 | Alison H Affinati |
| National Institutes of Health | DK123115 | Paul Kievit |
| National Institutes of Health | P51 OD-11092 | Paul Kievit |
| American Diabetes Association | 1-19-PDF-09 | Paul V Sabatini |

The funders had no role in study design, data collection and interpretation, or the decision to submit the work for publication.

### Author contributions

Alison H Affinati, Conceptualization, Formal analysis, Investigation, Writing - original draft, Writing - review and editing; Paul V Sabatini, Conceptualization, Investigation, Writing - original draft, Writing - review and editing; Cadence True, Abigail J Tomlinson, Melissa Kirigiti, Sarah R Lindsley, Investigation, Writing - review and editing; Chien Li, Conceptualization, Supervision, Writing - review and editing; David P Olson, Paul Kievit, Resources, Methodology, Writing - review and editing; Martin G Myers, Conceptualization, Resources, Formal analysis, Supervision, Funding acquisition, Writing - original draft, Project administration, Writing - review and editing; Alan C Rupp, Conceptualization, Data curation, Formal analysis, Investigation, Writing - original draft, Writing - review and editing

### Author ORCIDs

Paul V Sabatini http://orcid.org/0000-0001-6613-566X
Martin G Myers https://orcid.org/0000-0001-9468-2046
Alan C Rupp https://orcid.org/0000-0001-5363-4494

### Ethics

Animal experimentation: All mice used in this study were maintained in accordance with University of Michigan Institutional Animal Care and Use Committee (IACUC), Association for the Assessment and Approval of Laboratory Animal Care (AAALAC) and National Institutes of Health (NIH) guidelines under protocol number PRO00007438 (PI Myers). Nonhuman primate tissue was obtained post-mortem from the Tissue Distribution Program at ONPRC. Animal care is in accordance with the recommendations described in the Guide for the Care and Use of Laboratory Animals of the National Institutes of Health and animal facilities at the Oregon National Primate Research Center (ONPRC) are accredited by the American Association for Accreditation of Laboratory Animal Care International. ONPRC does not provide protocol numbers for security reasons.

### Decision letter and Author response

Decision letter https://doi.org/10.7554/eLife.69065.sa1

## Additional files

### Supplementary files

• Source data 1. ESµ values from CELLEX for each conserved ventromedial hypothalamic nucleus (VMH) neuron cluster using macaque data.

• Source data 2. ESµ values from CELLEX for each conserved ventromedial hypothalamic nucleus (VMH) neuron cluster using mouse data.

• Supplementary file 1. Nr5a1$^{eGFP-L10a}$ ventromedial hypothalamic nucleus (VMH) translating ribosome affinity purification with RNA-sequencing (TRAP-seq) enrichment results.

• Supplementary file 2. Ventromedial hypothalamic nucleus (VMH) pseudo-translating ribosome affinity purification with RNA-sequencing (TRAP) enrichment results.

• Supplementary file 3. ESµ values from CELLEX for each mouse ventromedial hypothalamic nucleus (VMH) neuron cluster.

• Supplementary file 4. ESµ values from CELLEX for each mouse ventromedial hypothalamic nucleus (VMH) neuron class.

• Supplementary file 5. Lepr$^{Slc17a6}$-L10a ventromedial hypothalamic nucleus (VMH) translating ribosome affinity purification with RNA-sequencing (TRAP-seq) enrichment results.

• Supplementary file 6. Ventromedial hypothalamic nucleus (VMH) Lepr pseudo-translating ribosome affinity purification with RNA-sequencing (TRAP) enrichment results.

- Supplementary file 7. ESμ values from CELLEX for each macaque ventromedial hypothalamic nucleus (VMH) neuron cluster.
- Supplementary file 8. ESμ values from CELLEX for each conserved ventromedial hypothalamic nucleus (VMH) neuron cluster using combined data.
- Transparent reporting form

### Data availability

Sequencing data have been deposited in GEO under accession code GSE172207.

The following dataset was generated:

| Author(s) | Year | Dataset title | Dataset URL | Database and Identifier |
|-----------|------|---------------|-------------|------------------------|
| Myers MG | 2021 | Cross-Species Analysis Defines the Conservation of Anatomically-Segregated VMH Neuron Populations | https://www.ncbi.nlm.nih.gov/geo/query/acc.cgi?acc=GSE172207 | NCBI Gene Expression Omnibus, GSE172207 |

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
