## [Decision Letter]

**Acceptance summary:**

By comparing single-cell transcriptomic profiles of VMH neurons in mice and macaques, the authors identified several genetically- and anatomically distinct classes that are highly conserved. This cross-species atlas provides a foundation for future efforts to explore whether these neuronal classes regulate discrete aspects of behavior.

**Decision letter after peer review:**

Congratulations, we are pleased to inform you that your article, "Cross-Species Analysis Defines the Conservation of Anatomically-Segregated VMH Neuron Populations", has been accepted for publication in *eLife* as a resource article – rather than as a research article. Your article has been reviewed by 3 reviewers and the evaluation has been overseen by a Reviewing Editor and a Senior Editor.

The reviewers just request that data on Dlk1 is shown – you can subsequently add this as a supplementary figure. Please take note of the points below and we hope you will continue to support *eLife*.

*Reviewer #1:*

Genetic and neurogenetic manipulations support the idea that distinct subpopulations of VMH neurons regulate disparate functions, including energy balance, fight-or-flight responses, reproduction and aggression. An obstacle to more precise functional studies is the lack of unique molecular markers that can be used to target manipulations to a specific subpopulation. Several groups have used single cell transcriptomic approaches to parcellate VMH into dozens of clusters. Yet, due to interrelatedness between the groups, it has not been possible to define unique markers for individual subpopulations.

Rather than further subdividing the VMH into ever smaller subpopulations, Affenati, et al. used a cross-species approach to identify broad classes of neurons that, in theory, could each mediate a different type of function ascribed to the VMH. By comparing TRAP-seq and snRNA-seq data from mouse with snRNA-seq data from macaque, they identified 6 anatomically discrete "classes" of VMH neurons that are conserved across species and can be uniquely marked. Transcriptomic analyses of LepR VMH neurons (using a clever intersectional approach to capture LepR+ glutamatergic neurons) provide further support for the idea that they represent a distinct population of neurons. Finally, the demonstration that orthologous mouse and macaque VMH neuronal classes express similar sets of genes (although some specific markers are not conserved), supports the idea that they represent common cell types that regulate similar functions.

Strengths: This study generated a cross-species atlas of VMH neurons that provides a conceptual and technical foundation for future efforts to test the hypothesis that the neuronal classes identified here, in fact, regulate discrete aspects of behavior.

Weakness: While this parcellation scheme is conceptually innovative, genetic approaches have already been applied to study two of the largest neuronal classes identified here (LepR+ and Esr1+). Moreover, while Dlk1 is useful to uniquely identify VMH neurons, it should be noted that it broad expression in neighboring hypothalamic neurons limits its utility in genetic targeting strategies.

Dlk1 is broadly expressed in neighboring hypothalamic nuclei surrounding the VMH. While it is useful in the computation analyses performed here, limitations to the application of this marker in functional studies should be discussed.

*Reviewer #2:*

The ventromedial hypothalamus (VMH) is a brain area implicated in an array of behavioral, metabolic and physiological functions. The manuscript by Affiniti and colleagues uses TRAP-seq in mice, together with snRNA-seq in mice and macaque to identify and characterize populations of neurons within the VMH. From their analysis, they report that VMH neurons consist of 6 major classes, including two ventrolateral classes, three dorsomedial classes and a class of glutamatergic VMH neurons in the tuberal region. The studies are well-executed, the data of great interest, have translational relevance and are of value to the scientific community.

Using both TRAP-seq and snRNA-seq, VMH LepR populations were enriched in the genes Grpr149, Rai14 and Tnfrsf8. While these genes are expressed in the VMH-DM, similar to LepR, to further validate these findings, it would be advantageous to determine the % of these neurons that co-express LepR, and those that don't express LepR, although this is not required for publication.

Previous evidence suggests that the VMHvl contains two distinct subpopulations of Esr1+ neurons that are activated by different behavioral stimuli. These are expressed in different subdivisions of the vlVMH and preferentially project to different brain areas. The question, worthy of further discussion, is whether there are distinct subpopulations of VMH LepR neurons, and whether they project to different brain regions to mediate different biological effects. In support of this, recent work from the group suggests that activation of VMH LepR neurons reduces both food intake and increases energy expenditure.